# Investigation of the Use of Environmental Samples for the Detection of EHV-1 in the Stalls of Subclinical Shedders

**DOI:** 10.3390/v16071070

**Published:** 2024-07-03

**Authors:** Nicola Pusterla, Kaila Lawton, Samantha Barnum

**Affiliations:** Department of Medicine and Epidemiology, School of Veterinary Medicine, University of California, Davis, CA 95616, USA; kolawton@ucdavis.edu (K.L.); smmapes@ucdavis.edu (S.B.)

**Keywords:** EHV-1, subclinical shedder, environmental samples, qPCR

## Abstract

In populations of healthy show horses, the subclinical transmission and circulation of respiratory pathogens can lead to disease outbreaks. Due to recent outbreaks of equine herpesvirus-1 myeloencephalopathy (EHM) in the USA and Europe, many show organizers have instituted various biosecurity protocols such as individual horse testing, monitoring for early clinical disease and increasing hygiene and cleanliness protocols. The aim of this study was to determine the accuracy of detecting EHV-1 in the various environmental samples collected from the stalls of subclinical shedders. Four healthy adult horses were vaccinated intranasally with a modified-live EHV-1 vaccine in order to mimic subclinical shedding. Three additional horses served as non-vaccinated controls. All the horses were stabled in the same barn in individual stalls. Each vaccinated horse had nose-to-nose contact with at least one other horse. Prior to the vaccine administration, and daily thereafter for 10 days, various samples were collected, including a 6” rayon-tipped nasal swab, an environmental sponge, a cloth strip placed above the automatic waterer and an air sample. The various samples were processed for nucleic acid purification and analyzed for the presence of EHV-1 via quantitative PCR (qPCR). EHV-1 in nasal secretions was only detected in the vaccinated horses for 1–2 days post-vaccine administration. The environmental sponges tested EHV-1 qPCR-positive for 2–5 days (median 3.5 days) in the vaccinated horses and 1 day for a single control horse. EHV-1 was detected by qPCR in stall strips from three out of four vaccinated horses and from two out of three controls for only one day. EHV-1 qPCR-positive air samples were only detected in three out of four vaccinated horses for one single day. For the vaccinated horses, a total of 25% of the nasal swabs, 35% of the environmental stall sponges, 7.5% of the strips and 7.5% of the air samples tested qPCR positive for EHV-1 during the 10 study days. When monitoring the subclinical EHV-1 shedders, the collection and testing of the environmental sponges were able to detect EHV-1 in the environment with greater frequency as compared to nasal swabs, stationary strips and air samples.

## 1. Introduction

One of the greatest challenges of mitigating viral respiratory outbreaks in large horse populations is the ability to capture the silent and clinical spread of highly contagious viruses such as EIV, EHV-1 and EHV-4. Many large equestrian facilities and show venues have instituted protocols with the goal of recognizing early clinical disease through the routine assessment of rectal temperatures. However, such strategies do not consider silent shedding and the transmission of respiratory viruses, especially in a population of adult and often vaccinated horses. Various protocols have recently focused on either monitoring high-risk horses through the testing of nasal secretions or testing the environment for targeted infectious pathogens [1,2,3,4]. The first approach gives an insight into the shedding status of horses in real-time but does not allow for predicting future shedding. The targeted testing of nasal secretions is often difficult and impacted by the unwillingness of owners to enroll their horses in such a protocol, as well as costs and the overall sustainability. However, the testing of high-risk horses is still a valuable option during emergency situations such as outbreaks. Environmental monitoring has gained interest as the testing of high-traffic areas (i.e., wash racks or housing areas where horses spend most of their time) gives a temporal insight into pathogen buildup. While the latter strategy has shown great promise for monitoring environmental spread and identifying clusters (adjacent stalls that test positive for a specific respiratory pathogen) as a reflection of silent transmission, it has remained time-consuming to perform and costly to implement. As devastating as the COVID-19 pandemic has been for the global human population, it has brought up novel strategies for monitoring the presence and spread of SARS-CoV-2 [5,6]. Various studies have shown the successful detection of SARS-CoV-2 in the air of indoor and outdoor facilities housing symptomatic and asymptomatic COVID-19 patients [7,8]. Therefore, the aim of this study was to determine the accuracy of detecting EHV-1 in various environmental and air samples collected from the stalls of subclinical shedders. 

## 2. Materials and Methods

### 2.1. Study Population

The study was performed in an eight-stall barn using seven healthy adult horses. The study population was composed of three mares and four geldings ages 5–16 years (median age 10 years). Four study horses were vaccinated intranasally with a modified-live EHV-1 vaccine (Rhinomune, Boehringer Ingelheim Animal Health, St. Joseph, MO, USA) in order to mimic nasal viral shedding similar to subclinically infected horses [9,10]. The administration of the EHV-1 vaccine has been shown to reliably induce EHV-1 shedding for up to 5 days at amounts similar to what is expected in naturally infected horses. Three additional horses served as non-vaccinated controls. All horses were stabled in the same barn in individual stalls with four horses on one side of the isle and three horses on the other side (Figure 1). Each control horse was stabled next to a vaccinated horse and each horse had nose-to-nose contact with at least one other horse through grilled side walls. The horses were fed twice daily and had ad libitum access to water. Furthermore, the stalls were cleaned once daily and dirty bedding as well as manure were removed and replaced with fresh shavings. 

### 2.2. Sample Collection and Testing

Prior to vaccine administration and daily thereafter for a total of 10 days, the following daily samples were collected from every study horse: a 6” rayon-tipped nasal swab (Puritan^®^, Guilford, ME, USA), an environmental sponge (3M, St. Paul, MN, USA), a soft fabric cloth strip placed above the automatic waterer (Medipore H soft cloth tape, 3M, St. Paul, MN, USA) and an air sample. The nasal swabs were collected from the left and right rostral nasal passages by gently introducing the swabs along the ventral meatus and rotating them for 5 s. The biocide-free cellulose sponges measure 1.5 × 3 inches, are mounted at one end of a stick and are prehydrated with a neutralizing buffer diluent for the collection of the samples. Each stall was swabbed along the front corner where the food and water buckets were kept, the inside of the stall door and the front bars of the stall that faced the barn isle for a total surface of approximately 16 square feet. A total of ten 4 × 4-inch cloth strips were placed around the automatic waterer and randomly assigned a number from 1–10, corresponding to the daily strip to be collected. Last but not least, air sampling was performed using a commercial Coriolis Compact air sampler (Bertin Instruments, Rockville, MD, USA). The Coriolis Compact is a dry cyclonic collector intended for microbial air monitoring. Its innovative dry cyclonic technology aspirates the particles with an airflow rate of 50 L per minute and centrifuges them in a disposable cone. The collection time lasted 8 min per stall, keeping the instrument 12 inches from each study horse and also walking the stall with the instrument to collect aerosolized dust. The instrument was cleaned between collections to prevent carryover contamination. All procedures were approved by the Institutional Animal Care and Use Committee of the University of California. 

The various samples were processed for nucleic acid purification using an automated nucleic acid extraction system (QIAcube HT, Qiagen, Valencia, CA, USA) according to the manufacturer’s recommendations. Thereafter, the purified nucleic acids were tested for the presence of EHV-1 using a previously validated qPCR assay [11]. 

The frequency of EHV-1 detection for the various samples from vaccinated and control horses was determined and compared. 

## 3. Results

Prior to vaccination, all samples collected from the seven study horses and their environment tested EHV-1 qPCR-negative (Table 1). Because of the expected short EHV-1 shedding time, the nasal swabs were only collected on the day of vaccination and for 96 h thereafter. EHV-1 qPCR-positive nasal secretions were identified in the vaccinated horses only for 1 to 2 days, with viral loads ranging from 1769 to 1,491,914 gB genes/million cells (median 919,121 gB genes/million cells). The environmental sponges tested EHV-1 qPCR-positive for 2–5 days (median 3.5 days) in vaccinated horses and for 1 day in a single control horse (horse 4). The environmental sponges tested EHV-1 qPCR-positive anywhere from day 2 to day 10 post-vaccination. EHV-1 was detected by qPCR in stall strips from three out of four vaccinated horses and from two out of three controls for only one day (days 2, 5 and 10 for vaccinated horses and days 2 and 7 for control horses). EHV-1 qPCR-positive air samples were only detected in three out of four vaccinated horses for one single day (days 4 (2 horses) and 5 post-vaccination). For the vaccinated horses, a total of 25% of the nasal swabs, 35% of the environmental stall sponges, 7.5% of the strips and 7.5% of the air samples tested qPCR positive for EHV-1 during the study period. For the non-vaccinated horses, a total of 0% of the nasal swabs, 2.5% of the environmental sponges, 5% of the strips and 0% of the air samples tested qPCR-positive for EHV-1 during the 10 study days. Overall, the environmental sponges were more reliable in detecting EHV-1 by qPCR in the environment of the subclinical shedders as compared to other environmental sample types.

## 4. Discussion

The study results show that the horses vaccinated with a modified-live EHV-1 vaccine intranasally shed for a short period and, similar to subclinically infected horses, were able to contaminate the environment. Among a variety of samples collected from the environment of these so-called subclinical shedders, EHV-1 was most frequently detected in the environmental sponges as compared to the other types of environmental samples such as the strips and air samples. 

To bypass the challenges of dealing with contagious EHV-1 and being able to reproducibly induce subclinical shedding, an EHV-1 MLV intranasal vaccine protocol was used. This protocol has been successfully used in the past to study the systemic and mucosal antibody response post vaccine administration [9,10]. Historically, the vaccine-derived EHV-1 could be detected in the nasal secretions of vaccinated horses for 1–5 days. In the present study, only the EHV-1-vaccinated horses had molecular evidence of EHV-1 in nasal secretions, with peak levels similar to the ones seen in naturally infected subclinical shedders [12]. Furthermore, there was no evidence of EHV-1 transmission between the vaccinated and control horses, based on the lack of EHV-1 detection in the nasal secretions of the control horses throughout the study period. This observation relates to the low infectious nature of the EHV-1 vaccine strain. 

Among the three different environmental samples, the stall sponges showed the highest detection rate for EHV-1. Stall sponges have been used in recent years to monitor the environment of at-risk horses and have been shown to be more reliable for detecting respiratory pathogens as compared to nasal secretions [2,3,4]. While the testing of respiratory secretions in healthy sport horses gives a real-time insight into their shedding status, testing the environment determines the accumulation of respiratory pathogens over time. 

When swabbing a stall, it is important to focus on areas where the horse spends time, such as feeding areas. While most vaccinated horses shed EHV-1 for 1 day, the short shedding time and high amount of virus shed were enough to contaminate the environment allowing the detection of EHV-1 from day 2 to 10. While all vaccinated horses had EHV-1 qPCR-positive stall sponges, one non-vaccinated horse with direct contact to a vaccinated horse also had one single positive sponge. Because of the direct contact between the vaccinated and control horses through grill partitions, the positive EHV-1 sponge result from the control horse stall likely originated from the neighboring vaccinated horse. Because sport horses at equestrian events are often kept in stalls with solid walls and have no direct contact to neighboring horses, it is unlikely that a subclinical shedder would contaminate stalls beyond its own.

It is interesting to observe that stall sponges tested three times more frequently EHV-1 qPCR-positive as compared to stationary wipes placed over the automatic waterer, a place known to have frequent nose and mouth interactions. The difference may relate to the overall surface area collected, considering that the stall sponges collected material over a 16-square-foot area (2304 square inch), compared to the individual 16-square-inch wipes. Also, the arbitrarily chosen location above the waterer could have contributed to the lower EHV-1 detection rate. It is possible that a more frequented area, such as the feeding area, might have generated a higher detection rate of EHV-1. While wipes are less time-consuming to manage as compared to the collection of stall sponges, they might still represent a sound strategy for monitoring the environmental buildup of respiratory pathogens for high-risk horses at show venues. 

Surprisingly, the detection of EHV-1 in air samples was low and occurred in three out of four vaccinated horses around the middle of the study period. The detection of viruses in the air samples presents many challenges because viruses are present in the air only at extremely low concentrations, which translates to the necessity of sampling relatively larger air volumes. Furthermore, there are not yet standardized protocols on how to best collect aerosolized viruses in regard to the sampling distance from the target, height from the floor, flow rates and sampled air volumes. The portable dry cyclonic collection device used in the present study has been successful in monitoring outdoor and indoor spaces for the presence of SARS-CoV-2 [13,14]. In the latter studies, the air collection was performed over 30 min. The inability to detect EHV-1 around the peak shedding time may relate to the large size of the alphaherpesvirus, the lack of aerosolization or the collection protocol. It was of interest to note that EHV-1 was detected on the 4th and 5th day post-vaccination and likely reflects the environmental virus that was picked up during the dynamic collection phase while walking the stall around the horse and kicking up bedding in order to generate dust. More work is needed in order to maximize the air collection protocol, as such monitoring devices may be less time-consuming compared to environmental swabs. However, one of the drawbacks of air sampling and testing is the price of the air sampler and the disposable cones used to collect each air sample. While sponges (approximately USD 3.0/sponge) and strips (approximately USD 0.05/strip) are cheap disposable collection devices, the disposable cones for the air sample are more expensive (USD 13.0/cone). More studies are needed in order to determine if the convenient approach to sampling and testing air could replace the time-consuming collection of environmental stall samples, especially in situations where a high density of at-risk horses is housed in the same environment. 

## 5. Conclusions

In conclusion, the present data showed that, in an EHV-1 subclinical shedding model using a modified-live vaccine administered intranasally, the virus was able to be detected for a short period in nasal secretions. The environmental buildup of EHV-1 could be detected via various environmental samples collected daily over 10 days. Among the various samples, the environmental sponges were more reliable in detecting EHV-1 by qPCR in the environment of subclinical shedders as compared to the stationary strips and air samples. It is important to keep in mind that the goal of environmental monitoring for EHV-1 and other respiratory pathogens is to assess the buildup of such pathogens in the environment in order to act in case of clustering of positive stalls. 

## Figures and Tables

**Figure 1 viruses-16-01070-f001:**
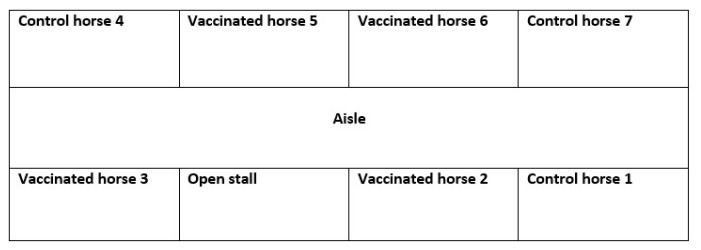
Layout of the barn housing four vaccinated horses and three unvaccinated controls. Four horses were stabled on one side of the aisle and three horses were stabled on the other side. One stall remained unoccupied.

**Table 1 viruses-16-01070-t001:** EHV-1 qPCR results listed as negative (white box) or positive (red box) for various biological and environmental samples collected from four vaccinated horses and three non-vaccinated controls. Gray boxes represent samples not taken.

Horse	Sample Type	Day 1	Day 2	Day 3	Day 4	Day 5	Day 6	Day 7	Day 8	Day 9	Day 10
**1 (control)**	**Nasal secretion**	negative	negative	negative	negative	negative					
	**Sponge**	negative	negative	negative	negative	negative	negative	negative	negative	negative	negative
	**Strip**	negative	positive	negative	negative	negative	negative	negative	negative	negative	negative
	**Air**	negative	negative	negative	negative	negative	negative	negative	negative	negative	negative
**2 (vaccinated)**	**Nasal secretion**	negative	positive	positive	negative	negative					
	**Sponge**	negative	positive	positive	positive	negative	negative	negative	negative	negative	negative
	**Strip**	negative	negative	negative	negative	negative	negative	negative	negative	negative	negative
	**Air**	negative	negative	negative	negative	positive	negative	negative	negative	negative	negative
**3 (vaccinated)**	**Nasal secretion**	negative	positive	negative	negative	negative					
	**Sponge**	negative	negative	negative	positive	positive	negative	positive	positive	negative	positive
	**Strip**	negative	negative	negative	negative	positive	negative	negative	negative	negative	negative
	**Air**	negative	negative	negative	negative	negative	negative	negative	negative	negative	negative
**4 (control)**	**Nasal secretion**	negative	negative	negative	negative	negative					
	**Sponge**	negative	positive	negative	negative	negative	negative	negative	negative	negative	negative
	**Strip**	negative	negative	negative	negative	negative	negative	positive	negative	negative	negative
	**Air**	negative	negative	negative	negative	negative	negative	negative	negative	negative	negative
**5 (vaccinated)**	**Nasal secretion**	negative	positive	negative	negative	negative					
	**Sponge**	negative	negative	negative	positive	positive	negative	negative	negative	negative	negative
	**Strip**	negative	positive	negative	negative	negative	negative	negative	negative	negative	negative
	**Air**	negative	negative	negative	positive	negative	negative	negative	negative	negative	negative
**6 (vaccinated)**	**Nasal secretion**	negative	positive	negative	negative	negative					
	**Sponge**	negative	positive	negative	negative	negative	positive	positive	positive	negative	negative
	**Strip**	negative	negative	negative	negative	negative	negative	negative	negative	negative	positive
	**Air**	negative	negative	negative	positive	negative	negative	negative	negative	negative	negative
**7 (control)**	**Nasal secretion**	negative	negative	negative	negative	negative					
	**Sponge**	negative	negative	negative	negative	negative	negative	negative	negative	negative	negative
	**Strip**	negative	negative	negative	negative	negative	negative	negative	negative	negative	negative
	**Air**	negative	negative	negative	negative	negative	negative	negative	negative	negative	negative

## Data Availability

The raw data supporting the conclusions of this article will be made available by the authors on request.

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
