# Peer review of "Investigation of the Use of Environmental Samples for the Detection of EHV-1 in the Stalls of Subclinical Shedders"

_viruses, 2024, doi:10.3390/v16071070_

Round 1
Reviewer 1 Report
Comments and Suggestions for Authors
The manuscript is a continuation of studies using environmental samples for detection of pathogens in horses. The purpose of the present study was to determine the accuracy of detecting EHV-1 in various environmental and air samples collected from the stalls of subclinical shedders which were horses inoculated with a live vaccine for mimicking subclinical infection. Experimental design is scientific and examined well. The authors showed that an environmental sponge sampling might be better than other samplings examined in the present study.
Specific comments are as follows.
L83-84: It is unclear what an environmental sponge is. The authors already reported what the environmental sponge was in references 2 to 4. Citing these references here might help readers understand this study.
L154-155: The authors described "there was no evidence of EHV-1 transmission between vaccinated and control horses. However, the authors did not show how they found no evidence of EHV-1 transmission between them. If no transmission occurred between them, the authors should show the evidence of no transmission between them, such as no detection of antibody responses in the control horses. Otherwise, it is unable to describe that there was no evidence of EHV-1 transmission between vaccinated and control horses.
L157-205: This paragraph is too long to read and should be divided into several paragraphs such as lines 157 to 162, 162 to 173, 173 to 180, 180 to 189, 189 to 193, 194 to 205.
Author Response
1. L83-84: It is unclear what an environmental sponge is. The authors already reported what the environmental sponge was in references 2 to 4. Citing these references here might help readers understand this study.
Additional information pertaining to the specific measurement and composition of the sponges was added under material and methods: "The biocide-free cellulose sponges measure 1.5 × 3 inches and are mounted at one end of a stick and prehydrated with neutralizing buffer diluent for the collection of samples".
2. L154-155: The authors described "there was no evidence of EHV-1 transmission between vaccinated and control horses. However, the authors did not show how they found no evidence of EHV-1 transmission between them. If no transmission occurred between them, the authors should show the evidence of no transmission between them, such as no detection of antibody responses in the control horses. Otherwise, it is unable to describe that there was no evidence of EHV-1 transmission between vaccinated and control horses.
The authors agree that the statement of "there was no evidence of EHV-1 transmission between vaccinated and control horses" needs further explanation. The statement is based on the lack of EHV-1 detection in nasal secretions collected from the control horses throughout the entire study period.
3. L157-205: This paragraph is too long to read and should be divided into several paragraphs such as lines 157 to 162, 162 to 173, 173 to 180, 180 to 189, 189 to 193, 194 to 205.
The authors agree with the reviewer regarding the length of the single paragraph. As suggested by the reviewer, the paragraph has been restructured in several smaller paragraphs.
Reviewer 2 Report
Comments and Suggestions for Authors
The objective of this research was to assess the accuracy of detecting EHV-1 in three types of environmental (environmental sponges, cloth strips and air samples) and biological samples (nasal swabs) obtained from the enclosures of asymptomatic shedders. EHV-1 was most frequently detected in environmental sponges when compared to the other types of environmental samples. The paper is well written, the cited references adequate and the experimental approach correctly designed.
The investigation was able to identify environmental sponges as a reliable sample to detect EHV-1 in a subclinical shedding context.
The presentation and discussion of DNA viral loads in positive samples are not properly presented. Nevertheless, this work does not fall within the scope of the study.
According to this reviewer, the manuscript contains information suitable for publication. However, it will be more appropriate to present the results as a brief report.

Author Response
Comment 1: The presentation and discussion of DNA viral loads in positive samples are not properly presented. Nevertheless, this work does not fall within the scope of the study.
The authors are not quite sure what the reviewer means with "DNA viral loads not properly presented". Various studies have shown that in order to properly quantitate viral loads using PCR, a standard curve for the target gene (in this instance the gB gene of EHV-1) as well as an internal housekeeping gene (equine glyceraldehyde-3-phosphate dehydrogenase) are required. In the present study, the CT values for both gB and eGAPDH were transformed into quantitative results based on the respective standard curve and the results were arbitrarily normalized as number of EHV-1 gB genes per million eukaryotic cells. Previous work on EHV-1 quantitation has shown that normalizing results against an internal control is more reliable than expressing the results per volume or amount of genomic DNA. Please let the authors know if this explanation is reasonable to retain the present data and interpretation or let us know how the quantitative results need to be reported.
Comment 2: According to this reviewer, the manuscript contains information suitable for publication. However, it will be more appropriate to present the results as a brief report.
Response: As suggested by the reviewer the type of article was switched from "Article" to "Brief Report".
Round 2
Reviewer 1 Report
Comments and Suggestions for Authors
The authors improved the manuscript well. The manuscript will provide us with new insights on EHV-1 infection research.